# Hamiltonian reconstruction as metric for variational studies

Kevin Zhang[1], Samuel Lederer[1], Kenny Choo[2], Titus Neupert[2],
Giuseppe Carleo[3] and Eun-Ah Kim[1]

**1** Laboratory of Atomic and Solid State Physics, Cornell University,
142 Sciences Drive, Ithaca NY 14853-2501, USA
**2** Department of Physics, University of Zurich,
Winterthurerstrasse 190, 8057 Zurich, Switzerland
**3** Computational Quantum Science Laboratory, École polytechnique fédérale de Lausanne,
Route Cantonale, 1015 Lausanne, Switzerland

## Abstract

Variational approaches are among the most powerful techniques to approximately solve quantum many-body problems. These encompass both variational states based on tensor or neural networks, and parameterized quantum circuits in variational quantum eigensolvers. However, self-consistent evaluation of the quality of variational wavefunctions is a notoriously hard task. Using a recently developed Hamiltonian reconstruction method, we propose a multi-faceted approach to evaluating the quality of neural-network based wavefunctions. Specifically, we consider convolutional neural network (CNN) and restricted Boltzmann machine (RBM) states trained on a square lattice spin-1/2 $J_1$–$J_2$ Heisenberg model. We find that the reconstructed Hamiltonians are typically less frustrated, and have easy-axis anisotropy near the high frustration point. In addition, the reconstructed Hamiltonians suppress quantum fluctuations in the large $J_2$ limit. Our results highlight the critical importance of the wavefunction's symmetry. Moreover, the multi-faceted insight from the Hamiltonian reconstruction reveals that a variational wave function can fail to capture the true ground state through suppression of quantum fluctuations.

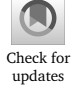

# 1   Introduction

The Hamiltonian is the defining object that governs the dynamics of a physical system. For a quantum mechanical system, it defines the Schrödinger equation to be solved to obtain the energy spectrum and the wavefunction. However, the approach of "exact diagonalization" is constrained to small system sizes due to the exponential growth of the Hilbert space upon increasing the system size. An alternative to exact diagonalization is the Quantum Monte Carlo technique using a stochastic approach to model the probability distribution associated with the thermal density matrix associated with a given Hamiltonian. These approaches, however, suffer from the sign-problem [1], which limits their applicability to a restricted class of Hamiltonians, or to high temperature properties only. These challenges motivated variational wavefunction approaches to start from many-body wave functions that are parameterized within a given functional form. In variational approaches, the Hamiltonian is referenced for optimizing the wavefunction within the chosen functional form (see the blue arrow in Figure 1). Since the resulting best wavefunction is constrained to lie within limited variational spaces such as tensor network states [2], neural network states [3, 4], and parametrized quantum circuits [5, 6] (see Figure 1), significant effort has been put into having sufficiently general variational classes that can capture the actual ground state. However, assessing how close a given variational parameterization is to the target ground state is, in general, a hard task.

At present, the standard metrics for assessing the quality of a wavefunction that cut across different variational forms are the energy and the energy variance. Reliance on these measurements, however, leaves the comparison between constructions a case-by-case trial exercise. Much needed are alternative metrics to assess the quality of a given variational state. Interestingly, recent works have proposed methods to reconstruct Hamiltonians from wavefunctions using measurements of correlators [7–9, 9–11] or single operator measurements [12, 13] (see the red arrow in Figure 1). These reconstruction processes have been tested on Hamiltonians with known exact solutions, but their applicability to challenging open problems has yet to be demonstrated.

In this Letter, we employ Hamiltonian reconstruction to investigate how frustration affects the bias (Figure 1) between reconstructed and target Hamiltonians for neural-network wavefunctions. We search for energy-minimizing wavefunctions in the space of convolutional neural network (CNN) and restricted Boltzmann machine (RBM) architectures, with the spin-1/2 $J_1$-$J_2$ Heisenberg model on a square lattice [4] as a target Hamiltonian. On this poster-child frustrated spin model, deep neural network-based wavefunctions have obtained highly

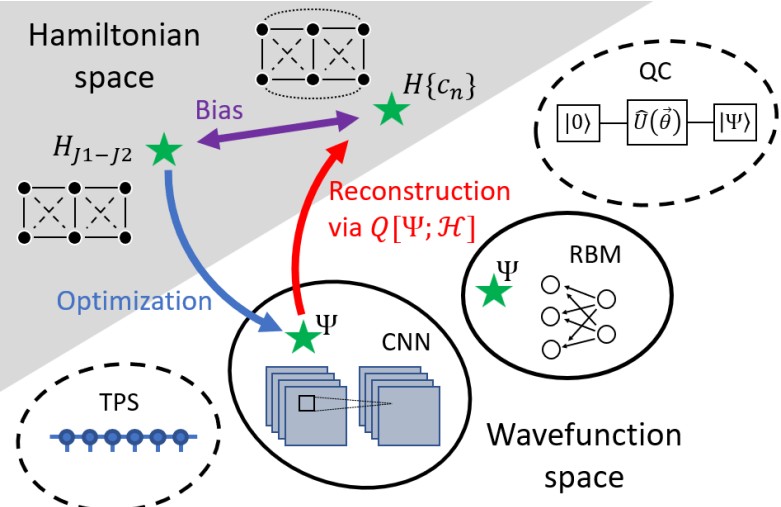

Figure 1: In a typical variational algorithm, a wavefunction is obtained through variational optimization within a given variational form such as CNN, RBM, tensor product state (TPS), or a parametrized quantum circuit (QC). In this work, we study CNN and RBM quantum states, marked with green stars. A blue arrow is shown to represent variational optimization of the CNN construction as an example. The Hamiltonian reconstruction works in the opposite direction to map a variational wavefunction to a Hamiltonian $H[\{c_n\}]$ (red arrow). The bias between the original Hamiltonian and the reconstructed Hamiltonian (purple arrow) provides insight into the nature of the variational wavefunction.

accurate results for the $J_1$-$J_2$ model away from the high frustration point, showing the potential of these variational constructions. However, these same states showed limitations near $J_2/J_1 = 0.5$, which is the point of high frustration [4]. To probe features of these wavefunctions, we construct subspaces of Hamiltonians that accommodate different "deformations" of the target Hamiltonian. For each subspace, we use the reconstruction method to retrieve the Hamiltonian that best fits the trained wavefunction. We then discuss insights from the reconstruction.

## 2 Hamiltonian reconstruction

For our goal of assessing variational wavefunctions, we chose to implement the approach of Refs. [7, 8]. The procedure starts with the wavefunction of interest $\Psi$, which is energy-optimized within a given variational form. We then define the Hamiltonian subspace to be searched by a spanning set of operators $\mathcal{O} \equiv \{O_i\}$. Any Hamiltonian that is an element of this subspace, i.e., $H \in \mathcal{H}$, can be expressed in the form

$$H[\{c_n\}] = \sum_n^{\dim(\{O_n\})} c_n O_n \,, \tag{1}$$

where $c_n$'s are real parameters. The aim of reconstruction is to find the $\dim(\{O_n\})$-dimensional vector $\{c_n\}$ such that the wavefunction of interest $|\Psi\rangle$ is most nearly an eigenstate of the corresponding Hamiltonian $H[\{c_n\}]$. For this, we construct the quantum covariance matrix $Q$ associated with the wavefunction and the Hamiltonian subspace

$$Q[\Psi; \mathcal{H}]_{nm} = \frac{1}{2} (\langle O_n O_m \rangle + \langle O_m O_n \rangle) - \langle O_n \rangle \langle O_m \rangle \,, \tag{2}$$

which is a $\dim(\{O_n\}) \times \dim(\{O_n\})$ positive semi-definite matrix where expectation values are evaluated with respect to the wavefunction $|\Psi\rangle$ (also see Figure 1). The number of expectation values to be measured for $Q$ is quadratic in the number of operators $\dim(\{O_n\})$, and therefore quadratic in the system size[1].

Hamiltonians that correspond to eigenvectors of $Q[\Psi; \mathcal{H}]$ with small eigenvalues would all accept $|\Psi\rangle$ as an approximate eigenstate. To see this, note that the variance of the Hamiltonian $H[\{c_n\}]$ in the state $|\Psi\rangle$ is given by

$$
\begin{aligned}
\langle (\Delta H[\{c_n\}])^2 \rangle &= \langle H[\{c_n\}]^2 \rangle - \langle H[\{c_n\}] \rangle^2 \\
&= \sum_{nm} c_n c_m \left( \langle O_n O_m \rangle - \langle O_n \rangle \langle O_m \rangle \right) \\
&= \vec{c}^{\,T} Q[\Psi; \mathcal{H}] \vec{c} \,.
\end{aligned}
\tag{3}
$$

By diagonalizing $Q[\Psi; \mathcal{H}]$, the Hamiltonians $H[\{c_n\}]$ which have the lowest variance under $|\Psi\rangle$ can be found, and the associated eigenvalues will be the variances of those Hamiltonians. If $|\Psi\rangle$ is an exact ground state of the original parent Hamiltonian $H^*$, and $H^*$ is within the Hamiltonian search space $\mathcal{H}[\mathcal{O}]$, then $H^*$ will lie in the nullspace of $Q[\Psi; \mathcal{H}]$.

The expectation values of many-body operators in Eq. (2) need to be evaluated by performing high-dimensional integrals. Typically, these integrals can be approximated via Monte Carlo (MC) sampling, but we found that the Hamiltonian reconstruction is sensitive to noise in the correlation functions (see Appendix B.2). This sensitivity restricts the procedure to systems where the correlation functions can be evaluated accurately. Indeed, previous applications of Hamiltonian reconstruction [8,14] have only treated well-understood states in which correlation functions can be evaluated exactly. In our case, this restricted our study to small system sizes in which the correlation functions could be evaluated explicitly.

The antiferromagnetic $J_1$-$J_2$ model for spin $1/2$ [15,16] is defined by the following Hamiltonian

$$
H_{J_1 J_2} \equiv J_1 \sum_{\langle ij \rangle} \vec{S}_i \cdot \vec{S}_j + J_2 \sum_{\langle\langle ij \rangle\rangle} \vec{S}_i \cdot \vec{S}_j \,,
\tag{4}
$$

where $\langle ij \rangle$ and $\langle\langle ij \rangle\rangle$ denote nearest and next-nearest neighbours respectively. We set $J_1 = 1$ and consider antiferromagnetic interactions $J_2 \geq 0$ for the $4 \times 4$ periodic 2D square lattice. The exact ground states of the Hamiltonian in the limits $J_2 \ll J_1$ and $J_2 \gg J_1$ are well understood, since geometric frustration is absent in both limits: the ground state is a Néel antiferromagnet for $J_2 \ll J_1$ and a stripe antiferromagnet for $J_2 \gg J_1$. However, the nature of the ground state in the vicinity of the maximally frustrated point of $J_2/J_1 = 0.5$ is the subject of much debate [17–26].

## 3 Hamiltonian space and wavefunction space

We consider three Hamiltonian subspaces that allow the reconstructed Hamiltonian to deviate from the target Hamiltonian Eq. (4) in physically meaningful ways. We chose the three two-

---

[1]The number of local operators is linear in the system size, so the number of evaluated expectation values is quadratic. Also, the computational cost remains the same when assuming translation invariance and summing operators over the system, since $\dim(\{O_n\})$ is smaller, but the total number of terms in the product $\langle O_n O_m \rangle$ is larger to compensate.

operator parametrizations

$$H[\alpha] = H_{J_1 J_2} + \alpha \left( \sum_{\langle i,j \rangle} S_i^z S_j^z + \frac{J_2}{J_1} \sum_{\langle\langle i,j \rangle\rangle} S_i^z S_j^z \right),$$

$$H[\delta J_2] = H_{J_1 J_2} + \delta J_2 \sum_{\langle\langle i,j \rangle\rangle} \vec{S}_i \cdot \vec{S}_j, \qquad (5)$$

$$H[J_3] = H_{J_1 J_2} + J_3 \sum_{\langle i,j \rangle_3} \vec{S}_i \cdot \vec{S}_j,$$

where $\alpha$ represents easy-axis anisotropy, and $\delta J_2$ and $J_3$ modify the next-nearest neighbor and longer range spin couplings. The coefficients of the original $J_1$-$J_2$ Hamiltonian are normalized to 1. For each possible two-dimensional Hamiltonian space, we constructed the matrix $Q$ independently, allowing us to study effects of individual perturbations. However, we found the results of higher-dimensional reconstructions allowing simultaneous perturbations of $\alpha$ and $\delta J_2$ to be consistent with those of individual reconstructions; see Results below. We specify technical details of the reconstruction quality in Appendix B.

Seeking further understanding of the challenges underlying the maximally frustrated point, we focus on neural network based wavefunctions that outperformed (i.e., had lower energy than) leading variational constructions, away from the high frustration point [4]. Neural networks can be universal approximators of complex functions [27, 28] and thus have the potential to allow more efficient exploration of the wavefunction space compared to traditional constructions [29]. The initial proposal of using restricted Boltzmann machines (RBM) to represent many-body wavefunctions [3] generated much excitement and spurred extensive investigations of RBM-based wavefunctions and their variants [26, 30–40]. More recently, Y. Levine et al, [41] showed that the more expressive convolutional neural network (CNN) architecture can encode volume-law entangled states more efficiently. Indeed, CNN wavefunctions improved on energy compared to state-of-the-art methods for the $J_1$-$J_2$ model, but only in the parameter regime away from the high frustration point of $J_2/J_1 = 0.5$ [4].

In this work, we examine CNN and RBM many-body wavefunctions. Both architectures preserve the translational invariance of the system, and the wavefunctions were further symmetrized to respect time reversal and point group symmetries, i.e., translation and rotation. The wavefunctions were not symmetrized with respect to the $SU(2)$ spin rotation symmetry of the Hamiltonian. Our restriction to the total $S_z = 0$ sector entails a residual $U(1)$ in-plane rotation symmetry, permitting easy-axis anisotropy in our trained wavefunctions. We trained wavefunctions for values of $J_2$ ranging between 0 and 2, and their optimization was done using the NetKet package [42]. For implementation, training, and symmetrization details, see Appendix A.

We now present metrics for verifying the reconstructed Hamiltonians, focusing on the RBM wavefunctions for simplicity. By construction, the variance of the reconstructed Hamiltonian over the trial wavefunction must be less than or equal to the variance of the target Hamiltonian. In the highly frustrated region with significant reconstructed anisotropy $\alpha$ for the $H[\alpha, \delta J_2]$ parameter space (Figure 2(a)), we illustrate that the variance is improved by roughly 10% (Figure 2(b)) for values of $\alpha$ of order 0.1. However, the energy variance should not be the only metric for assessing a given wavefunction. We next compare the overlap of the trial wavefunctions to the exact ground states of the original and reconstructed Hamiltonians, where overlap is defined as

$$|\langle \Psi_{exact} | \Psi_{var} \rangle|^2. \qquad (6)$$

In the frustrated regime, the overlap of the trial state with the ground state of the reconstructed Hamiltonians is significantly higher than the overlap with that of the original Hamiltonian. These metrics demonstrate that the reconstructed Hamiltonians are better descriptors for the variational wavefunctions.

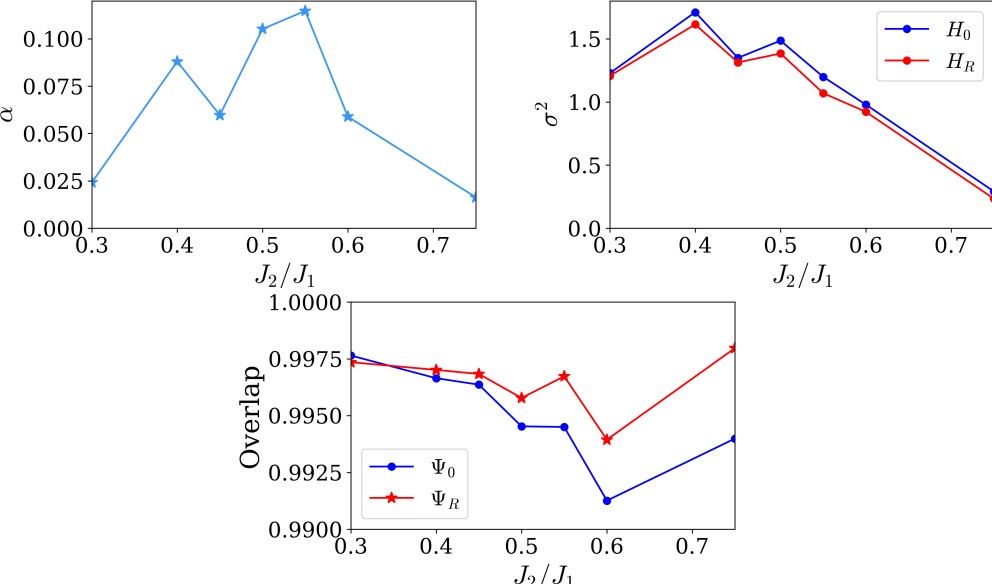

Figure 2: a) Reconstructed anisotropy $\alpha$ for RBM wavefunctions in the parameter region around the high frustration point. b) A comparison of the energy variance $\sigma^2 = \langle \Psi_{var} | \Delta H^2 | \Psi_{var} \rangle$ of original target Hamiltonian ($H_0$, blue line) to that of the reconstructed Hamiltonian in the space $H[\delta J_2]$ ($H_R$, red line). c) Overlap of the variational states with the exact ground states of the reconstructed Hamiltonians ($|\Psi_R\rangle$), compared with the overlap of the variational states with the exact ground states of the original Hamiltonians ($|\Psi_0\rangle$).

## 4 Results

The conventional measure for a wavefunction's quality is its variational energy. The energies of our trained wavefunctions, compared to the exact ground state energies, are shown in Figure 3(a); the high frustration region around $J_2 = 0.5$ is marked by a sharp peak in energy difference. The energy difference also remains large in the $J_2 > 0.5$ regime. The non-trivial dependence of the energies on the $J_2/J_1$ ratio implies multiple tendencies at play, yet no information is revealed about which factors affect the wavefunctions' performance for specific regions of parameter space. We therefore compare reconstruction results shown in Figure 3(b-d) to the variational energy to gain much needed insight.

The comparison between the reconstructed anisotropy $\alpha$ and the energy difference reveals two important features. The anisotropy is sharply peaked near $J_2/J_1 = 0.5$, indicating that anisotropy may hold the key to solving the high frustration point. This reinforces the importance of enforcing spin rotation symmetry on the wavefunctions, as was suggested by the performance of $SU(2)$ symmetric RBM wavefunctions for the 1D Heisenberg model [43]. However, the errors in energy away from the vicinity of $J_2/J_1 = 0.5$ arise from different sources. Also, despite significant energy differences between the RBM and CNN wavefunctions at the peak, they are on par with each other in terms of anisotropy. While $\alpha$ is consistent with a conventional measure of correlator anisotropy (see Appendix B.3), and is seemingly the most important barrier against solving the high frustration point, the comparison reveals that other factors might also be important.

The reconstructed interaction strengths $\delta J_2$ and $J_3$ present complementary information. They show deviations from the target Hamiltonian in two regions: near the high frustration point $J_2/J_1 = 0.5$, and the large $J_2$ region (see Figure 3(c-d)). In the vicinity of the high

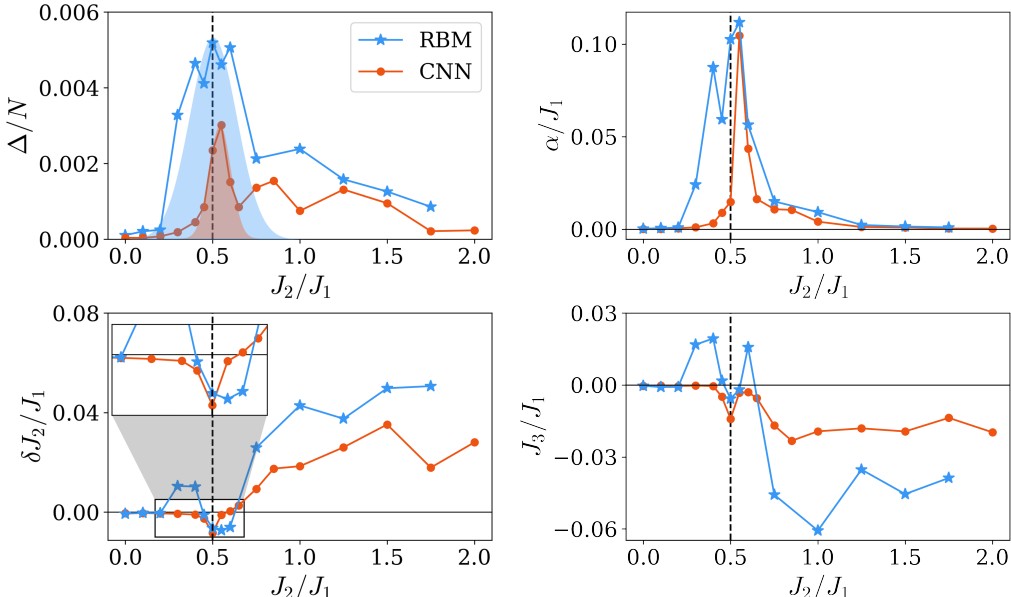

Figure 3: Various metrics for the CNN and RBM wavefunctions. The vertical broken line marks $J_2/J_1 = 0.5$, which is the high frustration point. a) Ground state energy difference (relative to exact ground state) per site. The shaded area is a guide to the eye that outlines the range in which the energy difference is attributable to the wavefunction anisotropy (see panel b). The unshaded energy difference in the large $J_2$ regime is associated with errors in the reconstructed spin couplings, panels c and d. b) The reconstructed easy-axis anisotropy $\alpha/J_1$, which is peaked at the classical high frustration point and tapers off in the small $J_2$ and large $J_2$ limits. c) The reconstructed difference in the nearest-neighbor coupling, $\delta J_2/J_1$. The reconstruction deviates from 0 around $J_2 = 0.5$, as well as in the large $J_2$ regime. d) The reconstructed longer-range interaction parameter $J_3/J_1$. Together with $\delta J_2/J_1$, these parameters are associated with the energy differences for large $J_2$.

frustration point, $\delta J_2$ tends to avoid $J_2/J_1 = 0.5$. This is shown by negative $\delta J_2$ below the high frustration point. In the large $J_2$ region, both $\delta J_2$ and $J_3$ imply strengthening of the stripe order and reduction of quantum fluctuations. Specifically, positive $\delta J_2$ and negative $J_3$ would both favor classical stripe order as we show explicitly in Appendix B.4. Successful implementation of SU(2) symmetry would likely resolve these issues by reflecting the true quantum fluctuations in the state, which could be the subject of future work.

Finally, we present results for a multi-dimensional reconstruction space in Figure 4 that shows agreement with the two-dimensional reconstructions. This space is defined by

$$
H[\delta J_2, \alpha] = H_{J_1 J_2} + \alpha \left( \sum_{\langle i,j \rangle} S_i^z S_j^z + \frac{J_2}{J_1} \sum_{\langle\langle i,j \rangle\rangle} S_i^z S_j^z \right) \\
+ \delta J_2 \sum_{\langle\langle i,j \rangle\rangle} \vec{S}_i \cdot \vec{S}_j \,.
\tag{7}
$$

Here we are allowing $\delta J_2$ and $\alpha$ to vary simultaneously[2]. Due to the larger dimensionality, the reconstructed Hamiltonians from this space will have lower variance than those obtained in two-dimensional reconstructions, and are therefore better parent Hamiltonians. Still, we find

---

[2]Including a $J_3$ term resulted in large reconstructed values of the parameters that are inconsistent with the reconstructed Hamiltonian being near the target Hamiltonian. In other words, the reconstructed parameters could not be interpreted as small perturbations. Hence, we did not include $J_3$ in the reconstruction.

that the results in Figure 4 are consistent with our results from individual two-dimensional spaces.

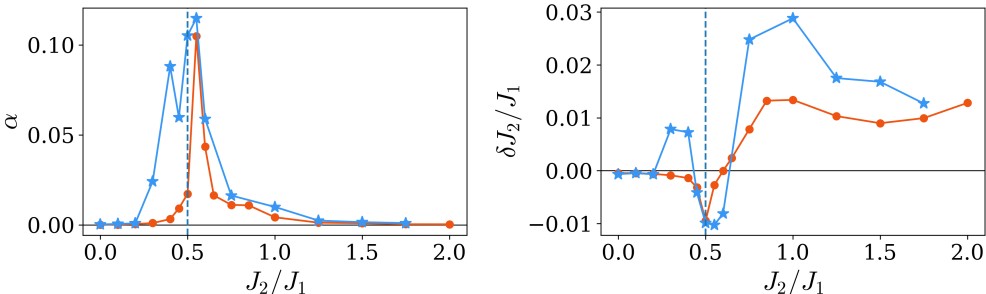

Figure 4: Results of the multidimensional reconstruction defined by Equation 7, with (a) reconstructed easy-axis anisotropy $\alpha/J_1$ and (b) difference in the nearest-neighbor coupling, $\delta J_2/J_1$. The trends in the parameters are consistent with our one-dimensional reconstructions in Figure 3.

We have compiled the results for the two-dimensional reconstructions of $\delta J_2$ and $J_3$ into one plot, Figure 5, to present a birds-eye view of the results. This demonstrates the tendency of the reconstructions to "push away from" the high frustration point, as well as suppress quantum fluctuations via ferromagnetic $J_3$ at large $J_2$. In other words, the reconstructions explaining the large energy differences for large $J_2$ (Figure 3(a)) can be summarized as a general tendency to suppress quantum fluctuations.

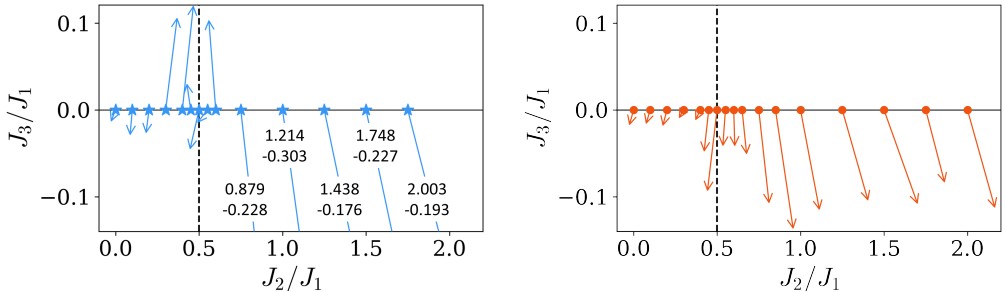

Figure 5: Schematic summary of $\delta J_2$ and $J_3$ reconstructions of (a) RBM and (b) CNN wavefunctions. The two reconstructions were performed separately, but here we have combined the two sets of results into one figure. The markers represent initial $J_2/J_1$ parameters for which we trained variational wavefunctions. The tips of the arrows show the *reconstructed* parameters, i.e., $(\delta J_2 + J_2)/J_1$ and $J_3/J_1$, with the deviations magnified by a factor of 5 for clarity. The annotations beside clipped arrows describe the locations of the arrowheads (upper: $J_2/J_1$, lower: $J_3/J_1$).

# 5   Conclusions

We have proposed Hamiltonian reconstruction as a method to probe many-body variational wavefunctions beyond their energies. Taking on the $J_1$-$J_2$ model and two neural network variational wavefunctions, RBM and CNN, we investigated the Hamiltonian spaces parametrized by three channels of deviations from the target model: $\alpha$, $\delta J_2$, and $J_3$. Our results dissect the $J_2/J_1$ parameter space into two regimes: the regime dominated by frustration ($J_2/J_1 \approx 0.5$) and the regime dominated by classical stripe order ($J_2/J_1 > 0.5$). We found the anisotropy $\alpha$

to be the dominant cause of error near the high-frustration point. Moreover, we found $\delta J_2$ and $J_3$ reconstruction to both indicate suppression of quantum fluctuation through artificial enhancement of classical order in the large $J_2$ regime. Overall, the Hamiltonian reconstruction revealed multiple ways for a variational wavefunction to fail in capturing highly frustrated ground states steeped in quantum fluctuations.

Looking ahead, we expect that Hamiltonian reconstruction can be an effective means to refine variational constructions in both classical and quantum (such as variational quantum eigensolver) platforms. With this method, specific areas of improvement for variational wavefunctions can be identified, informing future selection of variational constructions. Exploring reconstructions from wavefunctions of larger systems might also reveal new insights. Further, our results concerning the $J_1$-$J_2$ model may serve as guidelines for designing future neural network wavefunctions of similar frustrated spin systems.

# Acknowledgements

KZ, SL, and E-AK acknowledge NSF, Institutes for Data-Intensive Research in Science and Engineering – Frameworks (OAC-19347141934714). KC and TN acknowledge the European Research Council under the European Union's Horizon 2020 research and innovation program (ERC-StG-Neupert-757867-PARATOP).

# Appendix A   Neural network architectures and training

In this section, we will explain how we constructed and trained our variational ansatze. First, we describe a basic neural network to motivate the more complex architectures that we used. Then, we explicitly construct the convolutional neural network (CNN) and restricted Boltzmann machine (RBM) architectures that we used in our study. Finally, we explain how we trained and symmetrized the wavefunctions.

One of the simplest examples of a neural network that can be used to parametrize a variational wavefunction is a single-layer perceptron. Although we did not use this architecture, it serves as an illustrative example of neural network wavefunctions in general. The single-layer perceptron consists of one transformation (known as a dense layer)

$$w_i(\{v_{i'}\}) = g\left(\sum_j \mathbf{W}_{ij} v_j + b_i\right), \tag{8}$$

where $W$ is a complex matrix of tunable parameters ("weights"), $\vec{b}$ is a complex vector of tunable parameters ("biases"), and $g$ represents a non-linear firing function. The wavefunction corresponding to this single-layer network is

$$\langle\{\sigma_i\}|\Psi\rangle = \sum_i w_i(\{\sigma_{i'}\}). \tag{9}$$

Generically, a multi-layer perceptron can be constructed by composing several of these "linear" layers. Examples of common algorithms used to optimize these constructions are stochastic gradient descent or Adam. In our training, we used stochastic gradient descent with a learning rate of 0.001.

## A.1   Convolutional neural network

In a convolutional layer, both the input and output have an additional "channel" dimension on top of the spatial indices. Thus, each stage of intermediate values is a three-dimensional

tensor, with one channel index and two spatial indices. The input values (spin configuration) are interpreted as just consisting of a single channel. A convolutional layer from $N$ channels to $M$ channels is given by

$$w_{m,(i,j)}(\{v_{n,i',j'}\}) = g\left(\sum_{n}^{N}\sum_{x=0}^{L-1}\sum_{y=0}^{L-1} K_{n,(x,y)} v_{n,(i+x,j+y)} + b_m\right),\tag{10}$$

where the indices of the output vector $w$ are $m$ for the channel, and $(i,j)$ for the spatial position. The input vector $v$ is similarly indexed. As with the dense layer, $g(x) = \ln\cosh x$ is the nonlinearity. The $K$ parameters are called the "kernel" of the transformation, and there is one kernel for each output channel. Each kernel is a $L$ by $L$ by $N$ tensor of weights, which attempts to capture some spatially local feature of the input values. $L$ is the size of the kernel. The construction of the convolutional layer is explicitly translationally invariant, which respects the symmetry of the periodic $J_1$-$J_2$ model we are solving. The full convolutional network is given by the composition of convolutional layers,

$$\langle\sigma_{x,y}|\Psi\rangle = \sum_{w,i,j} w_{m,(i,j)}\left(\{v_{n,(i',j')}(...(\sigma_{x,y}))\}\right).\tag{11}$$

The outputs of the final layer were summed to obtain the output of the network. The network we used consists of three convolutional layers, with channel counts 6, 4, 2, and kernel sizes 4, 4, 2, respectively, amounting to a total of 236 parameters.

## A.2 Restricted Boltzmann machine

The single-layer restricted Boltzmann machine is characterized by the wavefunction

$$\langle\{\sigma_i\}|\Psi\rangle = e^{\sum_i a_i\sigma_i}\prod_{j}^{M}\cosh\left(\sum_i W_{ij}\sigma_i + b_j\right),\tag{12}$$

where $M$ represents the number of hidden nodes. We imposed the spatial symmetries of translation and rotation on the parameters $a_i$, $W_{ij}$, $b_j$, reducing the total number of free parameters by a factor of 16 (the size of the system). With a hidden node count of 160, this wavefunction has a total of 171 parameters.

## A.3 Sign rotation and symmetrization

Since the successful optimization of wavefunctions depends on a sign structure being pre-imposed, the wavefunctions were rotated in-plane:

$$|\Psi\rangle \rightarrow \left(\bigotimes_{i\in A} e^{-i\pi\sigma_z^i/2}\right)|\Psi\rangle,\tag{13}$$

where $A$ denotes a subset of lattice sites corresponding to either Néel or stripe order (Figure 6). The Néel rotation was used for $J_2 \leq 0.5$ and the stripe rotation used otherwise.

Finally, to ensure that the final wavefunctions respect point group symmetries and time reversal, we used the same procedure as in [4]. For an Abelian symmetry $C$, with an irrep $\Gamma$, the (unnormalized) symmetrization of the wavefunction $|\Psi\rangle$ under $\Gamma$ is given by

$$|\Psi_S\rangle = \sum_{r=0}^{|C|-1} \omega^r c^r |\Psi\rangle,\tag{14}$$

where $\omega$ is the character of $\Gamma$ and $c$ is the generator of $\Gamma$. For the $C_4$ rotation group, there are four possible irreps with characters $\pm 1$ and $\pm i$; for time reversal symmetry, there are two irreps with characters $\pm 1$. We chose the combination of representations that yielded the lowest ground state energy, and in all cases, the energy was lowest with the irrep with character 1 for both $C_4$ and $T$.

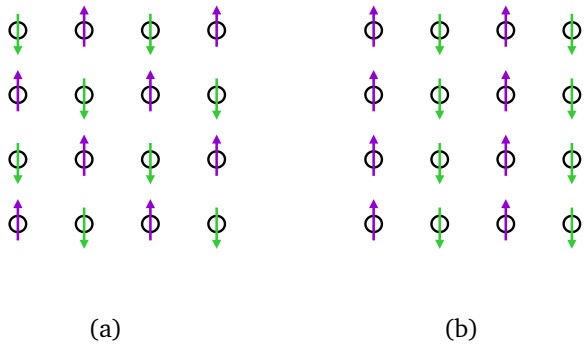

(a)                    (b)

Figure 6: Diagrams of the classical spin structure, used for in-plane wavefunction rotation, for a) Néel and b) stripe order.

## A.4   Consistency of optimization methods

We trained both architectures using stochastic reconfiguration (SR). We evaluated the loss function (i.e., the variational energy) of the wavefunctions using two different methods: exactly evaluated energy, and also Monte Carlo sampled energy. We also performed training with different sign rotations at the high frustration point. The reconstruction results with these different optimization schemes did not significantly vary, suggesting that the reconstructed biases are unlikely to be optimization artifacts for the neural network architectures we considered.

# Appendix B   Hamiltonian reconstruction

## B.1   Benchmarks

With the exact ground state of the Heisenberg model ($J_2 = 0$) obtained via exact diagonalization, Hamiltonian reconstruction indeed yields the correct Hamiltonian in all three subspaces we considered down to machine precision, as shown in Table 1. Because the exact ground state is used as the input state, reconstructions at any value of $J_2$ will yield the original Hamiltonian.

Table 1: Benchmark reconstructions for wavefunctions obtained via exact diagonalization for the $J_1$-$J_2$ model in the limit $J_2 = 0$. In all cases, the reconstruction yields the original $J_1$-$J_2$ Hamiltonian to within machine precision.

| $\mathcal{H}[\mathcal{O}]$ | Reconstructed parameter |
|---|---|
| $H[\delta J_2]$ | $\delta J_2 = -6.5 \times 10^{-15}$ |
| $H[J_3]$ | $J_3 = -3.18 \times 10^{-15}$ |
| $H[\alpha]$ | $\alpha = 3.73 \times 10^{-15}$ |

## B.2   Monte Carlo sampling

Various methods can be used when evaluating the elements of the quantum covariance matrix. Here, we present results of Hamiltonian reconstruction using Monte Carlo sampling for correlation functions. The model in question is the same 4×4 square lattice $J_1$-$J_2$ Heisenberg model, with $J_2/J_1 = 0.5$, i.e., the classical high frustration point. We only present results here for the reconstruction into the space $H[\delta J_2]$ using the convolutional neural network wavefunction.

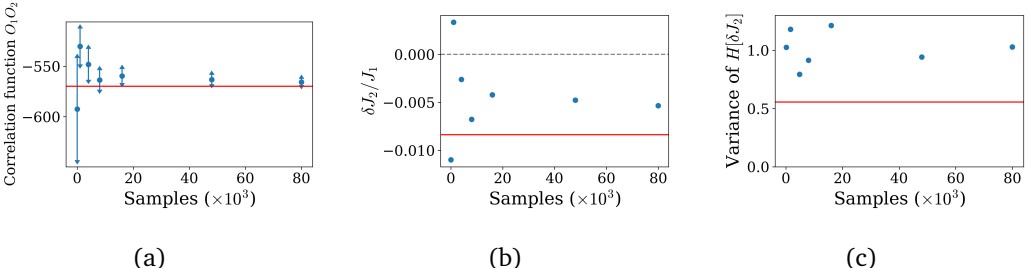

Figure 7: Results of Hamiltonian reconstruction using Monte Carlo sampling for operator correlation functions at $J_2/J_1 = 0.5$, into the space $H[\delta J_2]$. In all plots, the horizontal red line represents the true value as obtained through direct integration of the correlation functions. a) Correlation function $\langle H_{J_1 J_2} \sum_{\langle\langle i,j \rangle\rangle} \vec{S}_i \cdot \vec{S}_j \rangle$ as measured by Monte Carlo sampling. The error bars represent the standard deviation of the estimates for the correlation function. b) The reconstructed $\delta J_2$ parameter, which converges slower than the previous correlation function. c) The variance of the variational wavefunction $|\Psi\rangle$ under the reconstructed Hamiltonian. Here, the variance does not converge within the number of Monte Carlo samples that we used.

The reconstructed $\delta J_2$ parameter showed a slower rate of convergence (Figure 7(b)) than that of the shown correlation function (Figure 7(a)) with the number of Monte Carlo samples. Further, the variance of the input wavefunction under the reconstructed wavefunction did not converge for the range of Monte Carlo samples that we used (Figure 7(c)). Considering these points, we chose for our study to restrict our analysis to correlation functions evaluated explicitly. As a result, we limited ourselves to systems small enough that they were amenable to exact diagonalization.

### B.3   Comparison with anisotropy measure

In this section, we show that the $\alpha$ parameter is consistent with traditional measures of anisotropy. First, we present a "correlator anisotropy" measure, defined as

$$f(n) = 1 - \frac{\sum_{\langle ij \rangle_n} \left\langle S_i^z S_j^z \right\rangle}{\sum_{\langle ij \rangle_n} \frac{1}{2} \left\langle S_i^x S_j^x + S_i^y S_j^y \right\rangle} \, , \tag{15}$$

where $\langle ij \rangle_n$ represents a sum over $n$th nearest neighbors. As with the reconstructed $\alpha$, $f$ is sharply peaked near the high frustration point. We present in Figure 8 the anisotropy $f$ for various neighbor distances at $J_2 = 0.55$, which is where our reconstructed anisotropy $\alpha$ was maximal. Although $f$ and $\alpha$ did not completely agree at other $J_2$ values, at $J_2 = 0.55$ they consistently suggest that the CNN wavefunctions' anisotropy is somewhat less than that of the RBM wavefunctions in similar fashions. We conclude that although they are not identical measures of anisotropy, they are in agreement where the anisotropy is peaked.

### B.4   Spin-spin correlation functions of reconstructed Hamiltonians

Here we show how the spin-spin correlation functions support our conclusion that the variational wavefunctions are systematically less frustrated than the exact ground states. The spin-spin correlation function for wavevector $q$ is defined as

$$S(\vec{q}) = \frac{1}{N} \sum_{i,j} e^{i\vec{q}\cdot(\vec{r}_i - \vec{r}_j)} \langle \vec{S}_i \cdot \vec{S}_j \rangle \, , \tag{16}$$

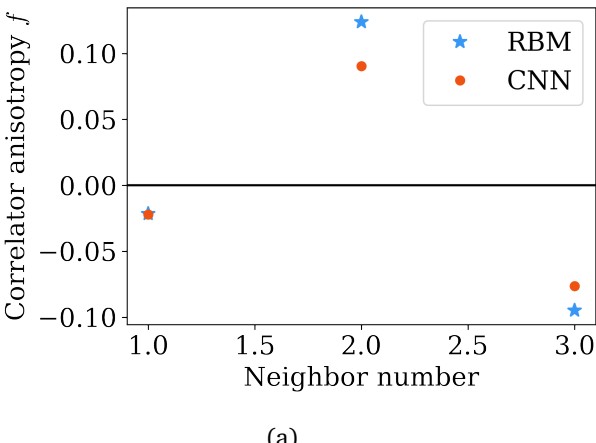

(a)

Figure 8: Correlator anisotropy $f$ (Equation 16) for RBM and CNN wavefunctions at various neighbor distances, at the point $J_2 = 0.55$ where our reconstructed anisotropy was peaked. The RBM anisotropies being closer to 0 than the CNN points is consistent with our reconstructed anisotropy $\alpha$, where the CNN wavefunction performed slightly better than the RBM.

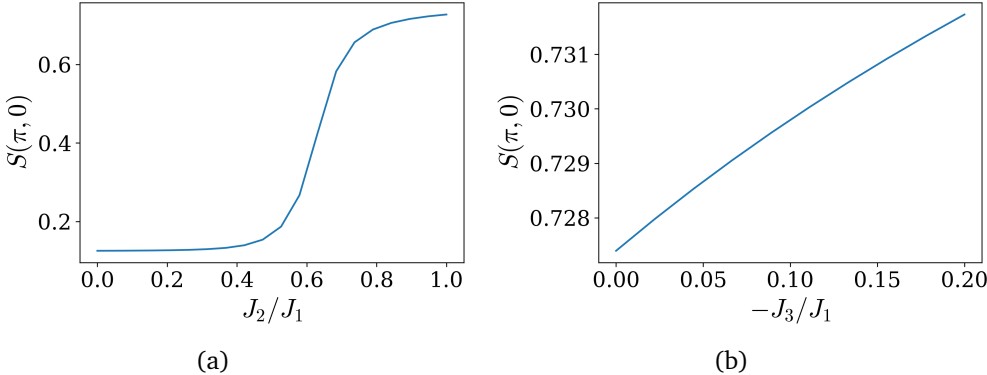

(a)                          (b)

Figure 9: Spin-spin correlation functions of the exact ground state of $J_1$-$J_2$-$J_3$ models. a) $S(\pi, 0)$ for $J_3 = 0$, showing how the stripe order is enhanced with increasing $J_2$. b) Similarly, at $J_2/J_1 = 1$, $S(\pi, 0)$ is enhanced with increasing ferromagnetic $J_3$.

where $N = 16$ is the system size. The correlation function evaluated at $\vec{q} = (\pi, \pi)$ is a measure for the strength of Néel order, while $\vec{q} = (\pi, 0)$ measures the strength of stripe order.

First, we show how the reconstructed $\delta J_2$, $J_3$ parameters tend to push the states towards less frustration, as shown by the spin-spin correlation functions $S(\vec{q})$ on the exact ground states of $J_1$-$J_2$-$J_3$ models. As can be seen from Figure 9, an increase in the $J_2$ parameter in the large $J_2$ regime leads to enhanced stripe order (Figure 9(a)). Also, small negative values of $J_3$ of similar magnitude to those reconstructed from our variational wavefunctions enhance the stripe order for states in the same regime (Figure 9(b)). The stripe order parameter here is the spin-spin correlation function with ordering vector $(\pi, 0)$ or $(0, \pi)$. These observations show that a ferromagnetic $J_3$ parameter, as well as positive $\delta J_2$, act to suppress quantum fluctuations and enhance the ground state order of the model in the large $J_2$ regime.

Next, we directly show that the variational states themselves are less frustrated than the exact ground states. We compared $S(\pi, 0)$ and $S(\pi, \pi)$ between the CNN wavefunctions and the exact ground states. Figure 10 shows that Néel correlations of the variational wavefunctions are stronger when $J_2/J_1 < 0.5$, while stripe correlations are stronger when $J_2/J_1 > 0.5$.

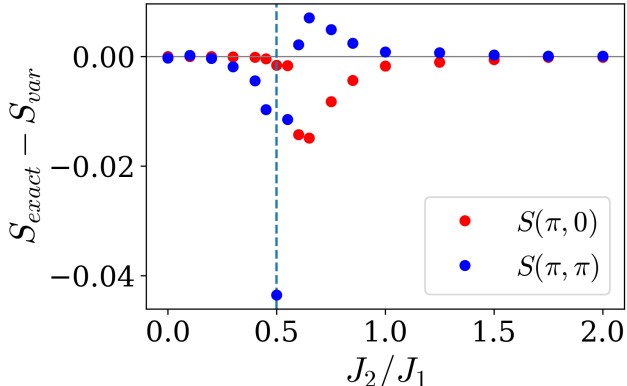

Figure 10: Difference between spin-spin correlation functions of CNN wavefunctions and exact wavefunctions, as a function of $J_2/J_1$. The variational wavefunctions have systematically stronger correlations, and are therefore less frustrated than the exact solutions to the $J_1$-$J_2$ Hamiltonian.

Since strong correlation functions are associated with the $J_2 \ll 1$ and $J_2 \gg 1$ limits of the model, these results show that the variational wavefunctions tend to be less frustrated than their exact counterparts.

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
