# Peer review of "Hamiltonian reconstruction as metric for variational studies"

_SciPost Physics, doi:SciPost Phys. 13, 063 (2022)_

## Round 1 · Referee Report · Anonymous · 2022-3-25

Strengths
1. Intriguing, promising, novel core idea: using the reconstruction of parent Hamiltonians to produce more interpretable analysis of many-body wavefunctions and ansatzes.
2. Clear and concise exposition.
Weaknesses
1. To see this idea to fruition, you would want to develop more "actionable" insights from their method, e.g. by using the analysis to develop new variational ansatzes. (Could be future work.)
Report
When an ansatz for a many-body wavefunction fails to capture the true groundstate, how can you understand what's "wrong" ? In what ways is the wavefunction produced by the ansatz deficient? You could evaluate the energy difference from the true groundstate energy, or the difference in correlation functions. But in general, it's hard to qualitatively understand many-body wavefunctions or how they differ from some target.
The authors tackle this problem creatively: given the estimated groundstate wavefunction, they try to understand its properties by asking which local parent Hamiltonian best matches it. The local parent Hamiltonian, unlike the wavefunction, then has more easily interpretable parameters.
This may be a promising approach for understanding many-body wavefunctions in general, including those produced by variational ansatzes as studied here.

---

## Round 1 · Referee Report · Anonymous · 2022-4-13

Strengths
1. The argument is well presented and the article reads effortlessly.
2. The authors tackle a general and important problem (evaluating the quality of a variational wave function as a candidate ground state for a given Hamiltonian), make an original contribution and bring promising leads.
3. The proposed analysis is quantitative and seems to give consistent results when varying the operator basis used for the reconstruction of the parent Hamiltonian.
Weaknesses
1. The article, apparently originally intended as a letter, would benefit from more detail on the points covered in the supplementary materials, especially section II. Taking advantage of SciPost's lack of size limitations, these details should fit naturally into the body of the article since they are part of the core message of this work.
2. Section IIC of Supplementary Material should be clarified/strengthened.
Report
In this article the authors tackle an important and general question in many-body physics : given a model Hamiltonian H whose ground state can only be described by a variational ansatz, how to estimate the validity of the state constructed to approximate the ground state of H ? The traditional approach is to put in competition variational states according to their energy, the lowest one being considered as the "best". The link between "low energy" and "correct description of correlations" is however far from being firmly established.
This work proposes a very original approach : (i) search for the hamitonian that would have as for its ground state the considered variational state and (ii) estimate how much this parent Hamiltonian differs from the one for which the variational state has been optimized, (iii) understand from (ii) what is missing in the trial wave wavefunction.
There is no doubt that this article provides enough new ideas, presented in a scientifically convincing manner, to deserve publication in SciPost. Figures 2, 3 and 4 give a very clear message and the conclusion brought by the authors that the CNN and RBM wavefunctions do not handle quantum fluctuations sufficiently to convincingly describe the maximally frustrated phase is well argued.
However, this article was apparently designed as a Letter, which justifies a number of important points being presented as supplementary material. But, since SciPost does not impose a length limit, some of the supplementary materials, especially section II which is very directly connected to the message of the article, would benefit from joining the body of the article, in a more detailed form. More specifically, further details would be appreciated to estimate the quality of the reconstruction itself. The discussion of this point is given as supplementary material II.C, but it raises some questions and would deserve a more extensive discussion.
To go into details one can ask by looking at figure 3a of section SM II.C why the variance of H_0 and H_R differ so little ? Would H_0 and H_R be finally as good parent Hamiltonians for \Psi_var ? If we consider the application sending Psi on the space of parent Hamiltonians, would it mean that it's almost stationary in the neighborhood of Psi ? In this case, can the small difference be really physically interpreted ? The same kind of questions arise on figures 3b and 3c (SM II.C.) : the variance around J2/J1=0.5 still seems large for H_R and the overlap with the exact wave function a bit disappointing when considering the rather modest size of the system (16 site cluster). In a word, is the operator space chosen for the reconstruction really appropriate ?
One last, minor point. The paper, starting with its title, brings the idea of a metric in the operator space which would allow to evaluate the distance between the initial Hamiltonian and the reconstructed Hamiltonian and hence the quality of the trial wave function as a "good" approximation of the ground state of the original Hamiltonian. This idea is again highlighted in a very suggestive way in figure 1. But I'm not sure we can really talk about "metric", as properly mathematically defined. Maybe "as a estimator for variational studies" would be more suitable.
Requested changes
Clarify/strengthen the argumenation in section "verification of reconstructed hamitlonian"
Incorporate part of Supplementary Materials (especially in sec. II) in the body of the article to make it self contained.

---

## Round 2 · Author Response

Dear Editor,

We thank the editor for organizing the review of our manuscript and are grateful to the referees for their valuable comments which have led us to improve the clarity of several key points of our work.
We are happy to see that all of the reviewers find our work to be of interest to the general community, and to be built upon a solid foundation.
In general, the referees hoped for more clarification for our claims.
We have taken this opportunity to address these concerns, and we believe that this has resulted in a much improved manuscript.

Below, we have compiled the referees' reports and our responses.
The changes to the text and Supplemental Material have been marked in red in the documents.

Referee 1
To see this idea to fruition, you would want to develop more "actionable" insights from their method, e.g. by using the analysis to develop new variational ansatzes. (Could be future work.)
-> We agree with the referee, and we have added a comment that developing the ansatz to improve the dJ2 or J3 reconstruction could be future work to the beginning of page 4.

Referee 2
Clarify/strengthen the argumenation in section "verification of reconstructed hamitlonian"
-> We have rewritten this section with more clarity on what our results imply.
Specifically, we have switched to the RBM wavefunctions (which more clearly display the improvement of the reconstructed Hamiltonians over the original, although the CNN wavefunctions also havethe same improvements). We show that the variance of the reconstructed Hamiltonian is improved by the same order of magnitude as the reconstructed Hamiltonian parameters, which is to be expected from the reconstruction space we use.
However, since the variance should not be used as the sole metric for wavefunction quality, as we argue in the article, we also show the improvement in wavefunction fidelity to the ground state by using the reconstructed Hamiltonian.

Incorporate part of Supplementary Materials (especially in sec. II) in the body of the article to make it self contained.
-> We agree with the referee that the move would help to make the article self contained, and we have done so accordingly.

---

## Round 2 · List of Changes

The changes to the text and Supplemental Material have been marked in red in the documents.

Specifically, the changes we have made to the manuscript are as follows:
- We have moved section II.C of the Supplemental Material to the main text in page 3 to make the text self-contained. We have also rewritten this section to improve the clarity and strength of our argumentation.
- We have added a comment that improving the ansatz to improve the dJ2 or J3 reconstruction could be future work to the beginning of page 4

You are currently on this page

Resubmission scipost_202202_00044v2 on 20 June 2022

---

## Editorial Decision

published